# Anchialine pool shrimp (*Halocaridina rubra*) as an indicator of sewage in coastal groundwater ecosystems on the island of Hawai'i

**Lisa C. Marrack**[1]*, **Sallie C. Beavers**[2]

**1** Department of Tropical Conservation Biology and Environmental Science, University of Hawaii, Hilo, Hawai'i, United States of America, **2** Kaloko-Honokōhau National Historical Park, Kailua-Kona, Hawai'i, United States of America

* lisamarrack@gmail.com

**Data Availability Statement:** All data files are available from the NPS IRMA DataStore: https://irma.nps.gov/DataStore/Reference/Profile/

## Abstract

Groundwater is a primary pathway for wastewater and other pollutants to enter coastal ecosystems worldwide. Sewage associated pathogens, pharmaceuticals, and other emerging contaminants pose potential risks to marine life and human health. Anchialine pool ecosystems and the endemic species they support are at risk and provide an opportunity to sample for presence of contaminants prior to diffusion in the marine environment. In this study, we tested the potential use of nitrogen isotopes in the tissues of a dominant anchialine pool grazing shrimp (*Halocaridina rubra*), as a bioindicator for sewage in groundwater flowing through their habitats. Water quality parameters and shrimp tissue isotopes (N and C) were collected from pools exposed to a range of sewage contamination along the West Hawai'i coastal corridor from 2015 to 2017. Data were used to test for spatial and temporal variability both within and among pools and to examine the relationship between stable isotopes and water quality parameters. Within 22 pools, mean $\delta^{15}N$ from whole tissue samples ranged between 2.74‰ and 22.46‰. Variability of isotope values was low within individual pools and within pool clusters. However, $\delta^{15}N$ differed significantly between areas and indicated that sewage is entering groundwater in some of the sampled locations. The significant positive relationship between $\delta^{15}N$ and dissolved nitrogen ($p<0.001$, $R^2 = 0.84$) and $\delta^{15}N$ and phosphorus ($p<0.001$, $R^2 = 0.9$) support this conclusion. In a mesocosm experiment, the nitrogen half-life for *H. rubra* tissue was estimated to be 20.4 days, demonstrating that the grazer provides a time-integrative sample compared to grab-sample measurements of dissolved nutrients. Ubiquitous grazers such as *H. rubra* may prove a useful and cost-effective method for $\delta^{15}N$ detection of sewage in conjunction with standard monitoring methods, enabling sampling of a large number of pools to establish and refine monitoring programs, especially because anchialine habitats typically support no macroalgae.

## Introduction

Wastewater derived inputs of nutrients and other contaminants have been documented as a source of degradation in lakes, rivers and coastal ecosystems worldwide [1, 2]. In coastal areas,

2299671. Marrack L and Beavers SC. 2023. Dataset for the project: Anchialine pool shrimp (Halocaridina rubra) as an indicator of sewage in coastal groundwater ecosystems on the island of Hawai'i. National Park Service. https://doi.org/10.57830/2299671.

**Funding:** The authors received no specific funding for this work.

**Competing interests:** The authors have declared that no competing interests exist.

nutrients, pathogens, pharmaceuticals, and other emerging contaminants from sewage pose potential risks to marine life and human health [3–5]. Over-enrichment in nitrogen (N) may drive reductions in water clarity, harmful algal blooms, hypoxia, and degradation of aquatic habitats including wetlands, coral reefs, and seagrass beds [6, 7]. Coastal areas throughout the world show signs of nutrient over-enrichment and growing human populations at the coast suggest that this trend may continue in the years ahead [8].

Groundwater is a primary pathway for wastewater contaminants to enter coastal aquatic ecosystems [9–11]. Globally, groundwater flow into the ocean, called submarine groundwater discharge (SGD), exceeds that of freshwater surface flow [10, 12]. As groundwater flows to the coast through permeable substrate such as karst or basalt, the water may entrain contaminants from onsite sewage disposal systems (OSDS) such as cesspools and septic tanks, storm drains, agricultural soils or other sources [13]. Submarine groundwater discharge is a documented pathway for sewage and other pollutants to enter nearshore habitats such as coral reefs [14, 15] and seagrass beds [16]. Prior to discharge in the ocean, wastewater contaminants may flow through groundwater-dependent habitats such as coastal wetlands and anchialine ecosystems [17, 18].

Nutrient sources from sewage have been successfully identified by examining the ratio of nitrogen stable isotopes ($^{15}N$: $^{14}N$) in water and organic tissue. Atmospheric $\delta^{15}N$ values range from 0‰ to 4‰ and synthetic fertilizer sources are typically more depleted with values from -2‰ to 4‰ [19, 20]. However, in sewage, $\delta^{15}N$ values are typically elevated (7‰ to 38‰) due to isotope fractionation during microbial metabolism [21]. Because algae incorporate nitrogen into their tissues from the surrounding water and have the same nitrogen isotope composition of that water, algal $\delta^{15}N$ has been used worldwide as an indicator of sewage in nearshore zones [16, 22–24]. Furthermore, because nitrogen isotopes transfer up the food web in a relatively predictable manner [25], invertebrates and fish have been successfully used as bioindicators to map sewage in some marine environments [26–28]. Due to metabolism and assimilation, $\delta^{15}N$ becomes enriched in herbivores compared to the plants they eat. On average, consumers are enriched by 3.4‰ compared to their food source via processes known collectively as trophic fractionation [29, 30]. For consumers raised on plants and algal diets, the enrichment has been measured as +2.2 ± 0.30‰ (SD) [31]. Therefore, grazer tissues at or above 8‰ to 9‰ may be an indication of sewage in a water body.

In Hawaiʻi, anchialine ecosystems provide an ideal location to sample exposed coastal groundwater prior to discharge and dilution in the marine environment [32]. Anchialine pools are groundwater-fed coastal habitats that are separated from the ocean by surface topography but are hydrologically connected underground [33]. Found throughout the world in porous basalt or karst substrate, these mixohaline habitats typically support endemic, rare, or listed species of crustaceans, mollusks, and other taxa [34]. Groundwater at the coast in leeward West Hawaiʻi is brackish and flows through anchialine pools out to the ocean [35, 36]. Researchers [37] sampled groundwater residence times in five West Hawaiʻi anchialine pools and measured residence at 1.7 to 5.5 hours with the shortest time during an ebbing tide. Furthermore, across 18 pools, pool flora and fauna did not appear to have a measurable effect on the concentration levels of total dissolved nitrogen entering or exiting the pools due to the rapid flow of groundwater through the pools [1]. Anchialine pool monitoring typically includes periodic grab sampling for dissolved nutrients [32]. However, given the apparent low residence time of groundwater in pools, nutrient or contaminant pulses may not be captured [38]. Stable isotope turnover rates may vary with species and tissue type [39], but offer a more time-integrated method to test for sewage inputs from groundwater than dissolved nutrient sampling alone.

The goal of this study was to test the potential use of the dominant anchialine pool grazer, the atyid shrimp *Halocaridina rubra*, as a bioindicator for sewage in groundwater flowing through coastal anchialine habitats. Adult *H. rubra* are usually less than 1.5 cm in length and may be found in densities as high as 175 per 0.25 m$^2$ feeding on biofilms that cover the rocky substrate [40, 41]. Healthy anchialine pools typically do not contain visible macroalgae or significant amounts of epilithon [42, 43], and therefore an alternate bioindicator organism would be useful. We hypothesized that as the dominant grazer of the system, *H. rubra*, could be used as a bioindicator of elevated $^{15}$N and sewage in groundwater. To test the suitability of *H. rubra* as a bioindicator we (1) evaluated the spatial patterns of nitrogen isotopes in *H. rubra* tissue at varying scales to see the relationship with known points of sewage influx; (2) examined the relationship between $^{15}$N in *H. rubra* tissue and various water quality parameters in a range of sewage influenced pools; and (3) examined *H. rubra* tissue turnover rates in a mesocosm experiment when shrimp were shifted from a low to a high $^{15}$N regime to enable correct interpretation of δ$^{15}$N levels in this primary consumer. Understanding the uptake and tissue turnover rates of nitrogen isotopes in taxa used as bioindicators may aid in comparisons between studies. This study represents the first effort in Hawai'i that the authors are aware of to examine wastewater contamination of groundwater using $^{15}$N levels in anchialine biota.

## Materials and methods

### Study site

Investigation into the use of *H. rubra* as a bioindicator of sewage contaminants in groundwater occurred along the western shore of the island of Hawai'i (West Hawai'i) where previous work has identified over 500 known anchialine pools along ~ 40 km of coastline [42–44]. The coastline of West Hawai'i is arid with rainfall ranging from 21 to 72 cm per year [45]. No perennial streams flow on the leeward side of the island [46]; therefore, groundwater is a primary pathway for land-based pollutants to enter the marine environment [17, 35, 47].

Anchialine pools were sampled along the West Hawai'i coastal corridor in areas with varying land-use. Focused research was conducted on a subset of pools found within Kaloko-Honokōhau National Historical Park ("park"), a 486-ha (1200-acre) national park unit with two large ancient Hawaiian fishponds, over 200 known anchialine pools, and extensive offshore coral reefs. The national park is surrounded by various urban land-uses that have the potential to impact groundwater quality including a large residential development and golf course on the northern boundary, upslope residential and light industrial development, and Honokohau Small Boat Harbor on the southern boundary (Fig 1) [18]. A disposal percolation basin for R-2 treated sewage is immediately upslope and to the south of the park near the Honokohau Harbor (Fig 1b). The National Park Service (NPS) has monitored groundwater quality on a quarterly basis since 2008 [32, 48]. These water quality data informed the sampling design of this study. The NPS and others' naming convention for Hawai'i Island pools is the first six letters of the land district followed by a number (e.g., Kaloko_10). Herein we shorten it to four letters (e.g., Kalo10; S1 Table).

### Spatial patterns of δ$^{15}$N in *H. rubra* tissue

An initial survey of δ$^{15}$N levels in *H. rubra* tissues was conducted in ten anchialine pools distributed along 40 km of coastline in West Hawai'i during August of 2015 (Fig 1). Seven to ten shrimp were sampled per pool, and stable isotope analysis was run for individual shrimp. Pools were in open spaces near a range of land-use types including undeveloped conservation areas, resorts, and urban areas. The number of On-Site Disposal Systems (OSDS) located

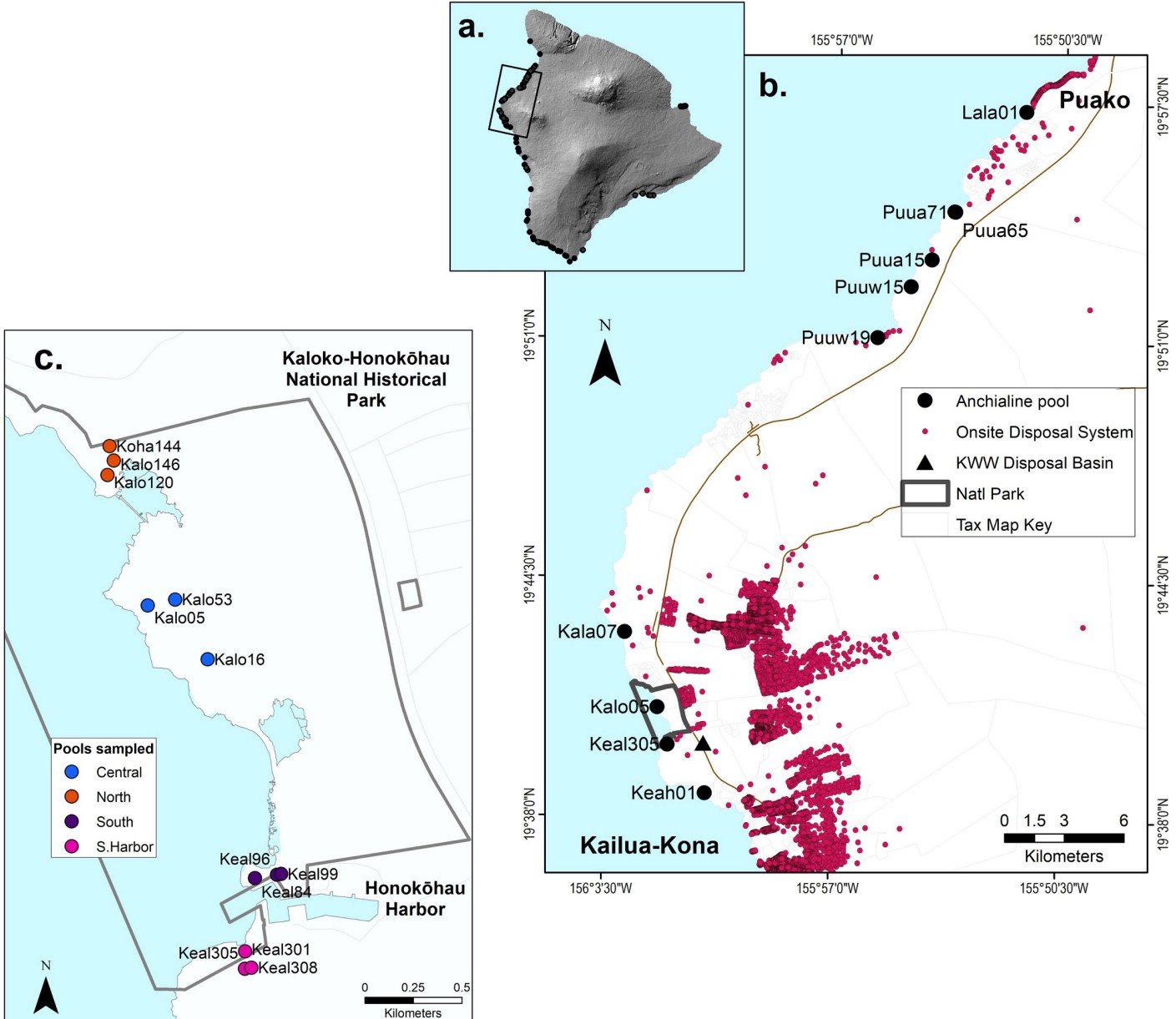

**Fig 1.** Maps of study area: (a) The Island of Hawaiʻi with the West Hawaiʻi study area outlined. Identified anchialine pool locations are indicated by black dots [50]; (b) West Hawaiʻi study area showing Kaloko-Honokōhau National Historical Park boundary and the Kealakehe Wastewater Disposal Percolation Basin (triangle). Anchialine pools sampled for *H. rubra* stable isotope analysis in 2015 are shown as large, closed circles. Onsite disposal systems (septic and cesspool) are shown as purple dots; (c) Pool locations within the national park sampled in 2016 and 2017 are shown as colored circles. The digital elevation model of the island of Hawaiʻi is a public domain GIS file from the USGS [49]. TMK boundaries, roads and onsite disposal systems are publicly available GIS files used with permission from the Hawaii State GIS program [50–52].

upslope from the pools varied from high density in the Kailua-Kona and Kaloko Honokōhau NHP area to low density in more northerly areas [53] (Fig 1).

Due to high levels of δ15N in pools south of Honokohau Harbor and a desire to further investigate isotope variability in a smaller spatial scale, we collected six to ten *H. rubra* by hand net from each of 12 pools within a 2-km section of coastline in Kaloko-Honokōhau NHP

during August 2016 and 2017 (Fig 1). Pool habitats ranged from bare lava with no microbial growth or terrestrial vegetation to those with some marine algae and/or plant canopy. To assess $\delta^{15}$N variability among pools that were in proximity to each other, clusters of three pools within 100 to 400 m of each other were sampled in four areas of the park (North, Central, South, South of Harbor). Sampling clusters of pools allowed testing of the assumption that pools in proximity have the same groundwater mass flowing through them and therefore similar nutrients and isotope signatures. In both 2016 and 2017, sampling was duplicated in all pools south and north of Honokohau Harbor and in the central portion of the park. At the north boundary (North), one pool had no visible shrimp (Kalo120) in 2016 and in 2017 a different pool (Kalo146) was dry; therefore, only one North pool (Koha144) was sampled both years. In all cases, shrimp were kept in cool water for two to four hours, then frozen for later analyses. Research within Kaloko-Honokōhau National Historical Park was conducted under NPS permits KAHO-2015-SCI-0008 and KAHO-2018-SCI-0008. All other locations were accessed with proper permission and collection methods were within guidelines approved by the State of Hawaii.

## Relationship between stable isotopes in *H. rubra* tissue and water quality parameters

We compared spatial and temporal variability of both stable isotope and water quality parameters within and among pools. To determine the relationship between nitrogen isotope levels in *H. rubra* and pool nutrient concentrations, we collected three replicate water samples and salinity and temperature measurements [29] concurrently with the shrimp collected during 2016 and 2017. The level of $\delta^{15}$N in shrimp at each pool was compared to salinity and to the concentrations of Nitrate-Nitrite ($NO_3^- + NO_2^-$), total dissolved phosphorus (TDP), and total dissolved nitrogen (TDN).

The $\delta^{15}$N data were also compared to the $\delta^{13}$C and C:N ratio measured within individual shrimp. Previous work in West Hawai'i showed that $\delta^{13}$C measurements of *H. rubra* reflect the diet available to them in the form of epilithon growing on rocks and subsidies from terrestrial vegetation [41]. Because dissolved nitrogen concentrations in pools have been found to depend on groundwater input rather than local pool processes [37], the expectation was that $\delta^{15}$N in shrimp tissues would not be correlated with $\delta^{13}$C levels. Likewise, C:N ratios were not expected to correlate to $\delta^{15}$N levels.

## Data analysis

To identify differences in shrimp-tissue $\delta^{15}$N values from a single pool between 2016 and 2017, pool means were compared using a paired T-test. No significant difference was found between years and data from the years were pooled. Differences between pool clusters within each sampling area were compared with a one-factor ANOVA. Post-hoc analyses used the Tukey HSD multiple comparisons test. Assumptions of normality and homogeneity of variance were tested prior to statistical analyses.

The relationship between shrimp tissue $\delta^{15}$N and mean nutrient concentrations by year were examined with linear regressions. The variability of water quality parameters within and among pools was examined across the two sampling years. All statistical analyses were performed in R [54] with *alpha* level set at 0.05.

## Isotopic turnover in *H. rubra* tissue

To identify the usefulness of *H. rubra* as a time-integrative indicator of sewage in groundwater, a determination of how quickly $\delta^{15}$N levels in shrimp tissues change relative to the isotope

levels in their diet is required. The relationship between the stable isotope composition of an animal's diet compared to its tissues relies on diet-tissue discrimination factors and tissue turnover times, which may vary by taxa, tissue type, and element [39, 55]. Stable isotope turnover rates in the tissues of consumers are expressed as half-lives, or the time for stable isotope values to reach 50% equilibrium with a new diet [39, 56]. Therefore, we conducted a 77-day diet shift experiment with *H. rubra* to quantify nitrogen incorporation rates. Approximately 100 *H. rubra* were hand-netted from a pool (Keal84) previously identified in this study as having low $\delta^{15}$N levels, i.e., no wastewater signal. This pool served as the control pool. Three shrimp were immediately collected to establish baseline $\delta^{15}$N. Over a 77-day period, shrimp were kept in glass mesocosms and fed a diet of biofilm from either a pool representative of a wastewater input (Keal301) or the control pool. Fifteen to 20 shrimp were placed in three replicates of each treatment. At one-week intervals, a single shrimp was collected from each mesocosm, air-dried, and frozen. At the conclusion of the experiment, all shrimp were oven dried for 72 h at 64°C and prepared for isotope analysis. After drying, each shrimp's total length to the nearest 1 mm and weight to the nearest 0.01 mg were recorded.

**Mesocosm design.** To provide shrimp with biofilm food sources and substrate that reflect anchialine pool habitat, sterilized, natural-lava gravel pillows were incubated in the high sewage or the control pool prior to distribution among the containers. Approximately six liters of natural lava gravel (3–5 cm diameter) were rinsed clean, soaked in a 10% bleach solution, rinsed, and oven-dried (sterilized) at 27.8°C. Gravel was then divided into six mesh bags constructed of black plastic shade cloth. For 20 days prior to the experiment, three bags were incubated in situ in the anchialine pool with a wastewater signal, and three labeled bags were incubated in the control anchialine pool. Gravel was changed weekly to ensure that the biofilm the *H. rubra* grazed on was fresh, plentiful, and retained the isotope levels observed in the specific pools examined. Each week, on a rotating basis, one bag from each pool was removed and divided between three mesocosms (~0.2 liters of gravel). Prior to adding fresh gravel, the gravel from the previous week was removed, rinsed thoroughly, re-bagged, and replaced in the appropriate pool for incubation.

Mesocosms were kept at equilibrium with air temperature (25°C) and were exposed to filtered natural light. Water for the mesocosms was collected from anchialine pools Keal301 (wastewater treatment) and Keal84 (control) and mesocosms were replenished on day 50. Three replicate water samples were collected for measurements of dissolved nutrients on day one and day 50. Similarly, replicate samples for dissolved nutrients were collected from the mesocosms on day 50 and at the end of the experiment (day 77).

Preparations for an approaching hurricane required the mesocosms be moved to an interior location under fluorescent lighting for approximately two days (day 63 to day 65). Additionally, the gravel substrate exchange did not occur on day 63 but occurred five days later. Therefore, no gravel exchange took place in week 10. All remaining mesocosm shrimp were collected for analysis in week 11.

**Isotopic turnover analysis.** Isotopic turnover was calculated as an exponential function of time following the diet-switch [57].

$$\delta_t = \delta_n + (\delta_o - \delta_n)e^{-(\lambda)t} \tag{1}$$

Where $\delta_t$ is the $\delta^{15}$N value of shrimp tissue at experimental time t, $\delta_o$ is the initial $\delta^{15}$N prior to the diet-switch, $\delta_n$ is the expected isotopic value for *H. rubra* in equilibrium with the new diet, and $\lambda$ is the fractional turnover rate per day. The value of $\delta_n$ was estimated using a non-linear regression. The mean $\delta^{15}$N of the three shrimp collected before the diet switch was used as the estimate of $\delta_o$ in the model. Using the experimental time (t) as the independent variable and

the corresponding $\delta^{15}N$ values of shrimp at time t ($\delta_t$) as the dependent variable, $\lambda$ was derived by fitting the exponential model in (Eq 1) to match the observed isotopic data. In this model, the turnover parameter ($\lambda$) combines growth of the animal and metabolic tissue turnover. Given the small size of *H. rubra*, turnover and nitrogen half-life in whole tissue were examined. The time needed to achieve the half-life, $T_{0.5}$ of nitrogen isotopes was calculated using methods from [57]:

$$T_{0.5} = \ln(0.5)/\lambda \tag{2}$$

## Stable isotope analysis

Isotopic signatures are calculated as $[(R_{sample} - R_{standard}) / R_{standard}] \times 1000$, where R = $^{15}N/^{14}N$ or $^{13}C/^{12}C$ and are expressed as $\delta^{15}N$ ‰ or $\delta^{13}C$ ‰ respectively [30]. Nitrogen and Carbon isotopes were analyzed by continuous flow triple isotope analysis using a CHNOS Elemental Analyzer interfaced to an IsoPrime100 mass spectrometer located at the Center for Stable Isotope Biogeochemistry at University of California at Berkeley. Long-term external precision for C and N isotope analyses within the Center are $\pm$ 0.10‰ and $\pm$ 0.20‰ respectively.

Prior to spectrometer analysis, tissue samples from the abdominal segment of individual shrimp were dried and weighed out in tin capsules. Samples collected for environmental gradient studies were freeze-dried prior to analysis. For the mesocosm experiment, shrimp were oven dried for 72 h at 64˚C.

One common method in stable isotope analysis of muscle tissue is lipid extraction prior to mass spectrometer analysis to correct for lipids in $\delta^{13}C$ values. Lipid extraction from estuarine and freshwater fish tissues has been found to cause an undesirable large shift in $\delta^{14}N$ values of (-2.11 to 2 per mil) [25]. Based on the small shift in $\delta^{13}C$ and large shift in $\delta^{14}N$ due to lipid correction, lipid extraction was not performed on tissue prior to analysis.

## Water quality analysis

Water samples were filtered through a pre-combusted (500˚C for 6 h) filter (GF/F Whatman™), and stored frozen until analysis for nutrient concentrations at the University of Hawai'i at Hilo Analytical Laboratory. Nutrients were analyzed on a Pulse Technicon™ II autoanalyzer using standard methods ($NO_3^-$ + $NO_2^-$ [Detection Limit (DL) 0.07 µmol/L, USEPA 353.4], total dissolved phosphorus (TDP) [DL 0.5 µmol/L, USGS I-4650-03], H4SiO4 [DL 1 µmol/L, USEPA 366]), and reference materials (NIST; HACH 307–49, 153–49, 14242–32, 194–49). Total dissolved nitrogen (TDN) was analyzed by high-temperature combustion, followed by chemiluminescent detection of nitric oxide (DL 5 µmol/L, ASTM D5176, Shimadzu TOC-V, TNM-1) [58]. Salinity, pH, and temperature were measured at the time of water collection using a YSI Pro 2030 multi-parameter probe.

## QA/QC

The quality assessment and control measures used in the water sampling portion of this study were developed by the National Park Service Pacific Island Network's water quality monitoring program [32]. These measures include calibrating the YSI sonde prior to field measurements, collecting water samples at an outgoing tide, collecting water samples in triplicate within previously acid-washed bottles at each pool, and freezing water samples prior to nutrient analysis. For the shrimp tissue samples, replicate samples were collected at each pool or mesocosm group within minutes of each other and were prepared as described in stable

isotope analysis above. Water nutrient level analyses and shrimp tissue isotope analyses were conducted at independent laboratories with their internal QA/QC protocols.

## Results

### Spatial patterns of δ¹⁵N in *H. rubra* tissues

In 2015, $\delta^{15}N$ values in *H. rubra* tissues varied among anchialine pools located along the West Hawai'i coastal corridor. Mean $\delta^{15}N$ values (± SD) ranged between 2.74 ± 0.37 ‰ and 22.46 ± 0.75 ‰ (Fig 2a). Eight of the pools had mean values below 8.9‰. The pools with the highest values were at Keal305 (22.46 ± 0.75 ‰), which is shoreward of the Kealakehe Wastewater Treatment Plant disposal percolation basin, and at Keah01 (9.59 ± 0.95 ‰) located near the ocean in Kailua-Kona. These elevated values indicate that human sewage inputs to groundwater are entering the anchialine food web.

Within the pools sampled in the national park in 2016 and 2017, mean $\delta^{15}N$ levels ranged from 3.91 ± 0.47 ‰ to 24.68 ± 0.68 ‰ (Fig 2b). In both years, the highest $\delta^{15}N$ levels were found in the three pools south of Honokohau Harbor (21.01‰ to 24.68‰) and were similar to levels measured in 2015 in the same area (Keal305: 22.46 ± 0.75 ‰). Shrimp in all other pools sampled in the park ranged between 2.74‰ and 10.89‰ (Table 1, Fig 2). The $\delta^{15}N$ levels showed no significant difference between sampling years 2016 and 2017 at individual pools (T-value (1, 9) = -0.2, p = 0.85). The $\delta^{15}N$ levels between geographic pool clusters within the park were significantly different (F-value (1,3) = 2007, p < 0.001) and each cluster was statistically different from the other three (p < 0.001). The Central area pool cluster had the lowest $\delta^{15}N$ levels (4.59 ± 0.66 ‰) while South Harbor pools had the highest levels (22.8 ± 1.31 ‰). Although the North cluster (7.59 ± 0.93 ‰) and South cluster (9.50 ± 1.31 ‰) $\delta^{15}N$ levels were significantly different, they showed the most overlap in $\delta^{15}N$ levels (Fig 2b).

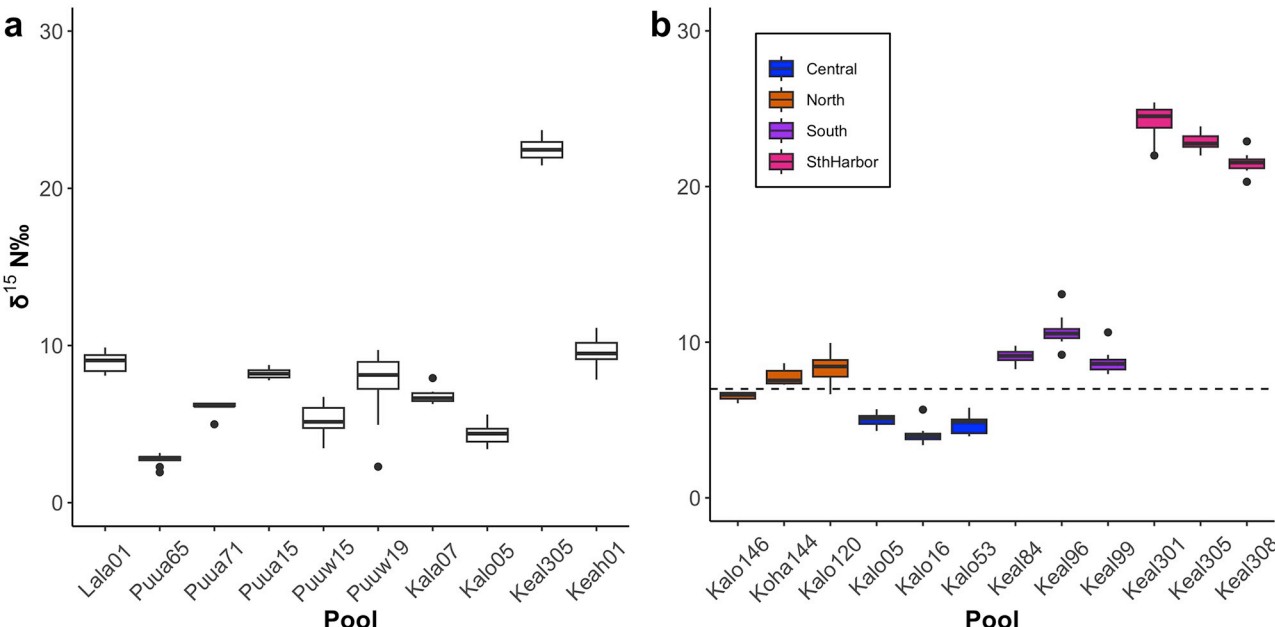

**Fig 2.** Variability of δ¹⁵N (‰) in *H. rubra* tissue sampled at individual anchialine pools: (a) Pools sampled along the West Hawai'i coastline are arranged from north to south (left to right); (b) Pools sampled 2016 and 2017 in Kaloko-Honokōhau National Historical Park. Colors indicate different pool locations (north boundary "North," central park "Central," south boundary "South," south of Honokohau Harbor "SthHarbor"). Box-and-whisker plots display the median, lower and upper quartiles.

**Table 1. Linear regression results examining the relationship between *H. rubra* tissue δ $^{15}$N, water quality parameters, C:N, and δ$^{13}$C in 2016 and 2017.** Nutrients are in mg/L and salinity in PSS.

| Relationship | Year | Linear equation | *p*-value of slope | R$^2$ |
|---|---|---|---|---|
| δ$^{15}$N to NO$_2$+NO$_3$ | 2016 | y = 3.4x + 4.1 | <0.001 | 0.83 |
| | 2017 | y = 7.96x + 2.5 | 0.04 | 0.33 |
| δ$^{15}$N to TDN | 2016 | y = 3.7x + 3.3 | <0.001 | 0.84 |
| | 2017 | y = 1.7x + 6.6 | <0.01 | 0.62 |
| δ$^{15}$N to TDP | 2016 | y = 5.3x + 14.6 | <0.001 | 0.9 |
| | 2017 | y = 5.9x + 13.7 | <0.001 | 0.9 |
| δ$^{15}$N to C:N | 2016 | y = -6.1x + 37.1 | 0.16 | 0.11 |
| | 2017 | y = -11.3x + 56.5 | 0.13 | 0.11 |
| δ$^{15}$N to δ$^{13}$C | 2016 | y = 1.4x + 36.9 | 0.08 | 0.22 |
| | 2017 | y = 1.2x + 35.5 | 0.15 | 0.13 |
| δ$^{15}$N to Salinity | 2016 | y = 0.6x + 2.1 | 0.33 | 0.01 |
| | 2017 | y = 0.7x + 0.99 | 0.26 | 0.04 |
| NO$_2$+NO$_3$ to Salinity | 2016 | y = 0.005x + 2.1 | 0.98 | -0.11 |
| | 2017 | y = -0.07x + 2.1 | 0.19 | 0.09 |

## Nutrients

Water quality nutrients (TDN, NO$_3$+NO$_2$, and TDP) varied between park pools and, in the case of TDN, varied among sampling dates (Fig 3, S2 Table). Throughout the park, the mean TDN concentrations measured within a single pool at any one time ranged from 6.04 ± 0.10 mg/L to 0.15 ± 0.00 mg/L, and mean TDP concentrations ranged from 1.26 ± 0.02 mg/L to 0.2 ± 0.00 mg/L. Nutrient concentrations were highest in the three South of Harbor pools with mean TDN values of 5.56 ± 0.5 mg/L in 2016 and 2.76 ± 0.11 mg/L in 2017 (Fig 3b). The highest observed TDP values were also in these southernmost pools for both 2016 (1.22 ± 0.05 mg/L) and 2017 (1.20 ± 0.04 mg/L) (Fig 3c). Within individual pools, TDP concentrations were more variable than TDP between the two sampling dates.

A significant positive linear relationship is evident between δ$^{15}$N levels in *H. rubra* tissue and nutrient concentrations in the water that the shrimp were collected from (TDN, NO$_3$+NO$_2$, and TDP) (Fig 3, Table 1). For TDP, the regression equations were—similar for 2016 (y = 5.3x + 14.6, p < 0.001, R$^2$ = 0.9) and 2017 (y = 5.9x + 13.7, p < 0.001, R$^2$ = 0.9). However, for TDN and NO$_3$+NO$_2$, the slopes differed between years (Table 1). This difference was primarily due to decreased TDN and NO$_3$+NO$_2$ values but the same levels of δ$^{15}$N within the pools south of the Harbor in 2017 compared to 2016.

Mean δ$^{13}$C values in shrimp tissue ranged from -13.76 ± 1.19 ‰ to -23.24 ± 1.34 ‰ throughout the park. No correlation between δ$^{15}$N and C:N of shrimp tissue is evident for either year (Fig 3d, Table 1). Likewise, no correlation is evident between δ$^{15}$N and δ$^{13}$C for either 2016 or 2017 (Fig 3d, Table 1).

During both sampling years, pool salinities ranged between 8.9 and 15.7 PSS except for a single pool (Keal84) that averaged 23.2 PSS. Salinities were relatively consistent between years. No correlation between salinity and δ$^{15}$N or TDN is observed for either year (Table 1).

## Isotopic turnover analysis

Shrimp fed by biofilms from pool Keal305 with expected sewage influence showed weekly increases of δ$^{15}$N in their tissues changing from an initial mean value of 8.99 ± 0.25 ‰ and ending with 17.16 ± 0.99 ‰ on day 77 (Fig 4). In contrast, shrimp fed biofilms from pool

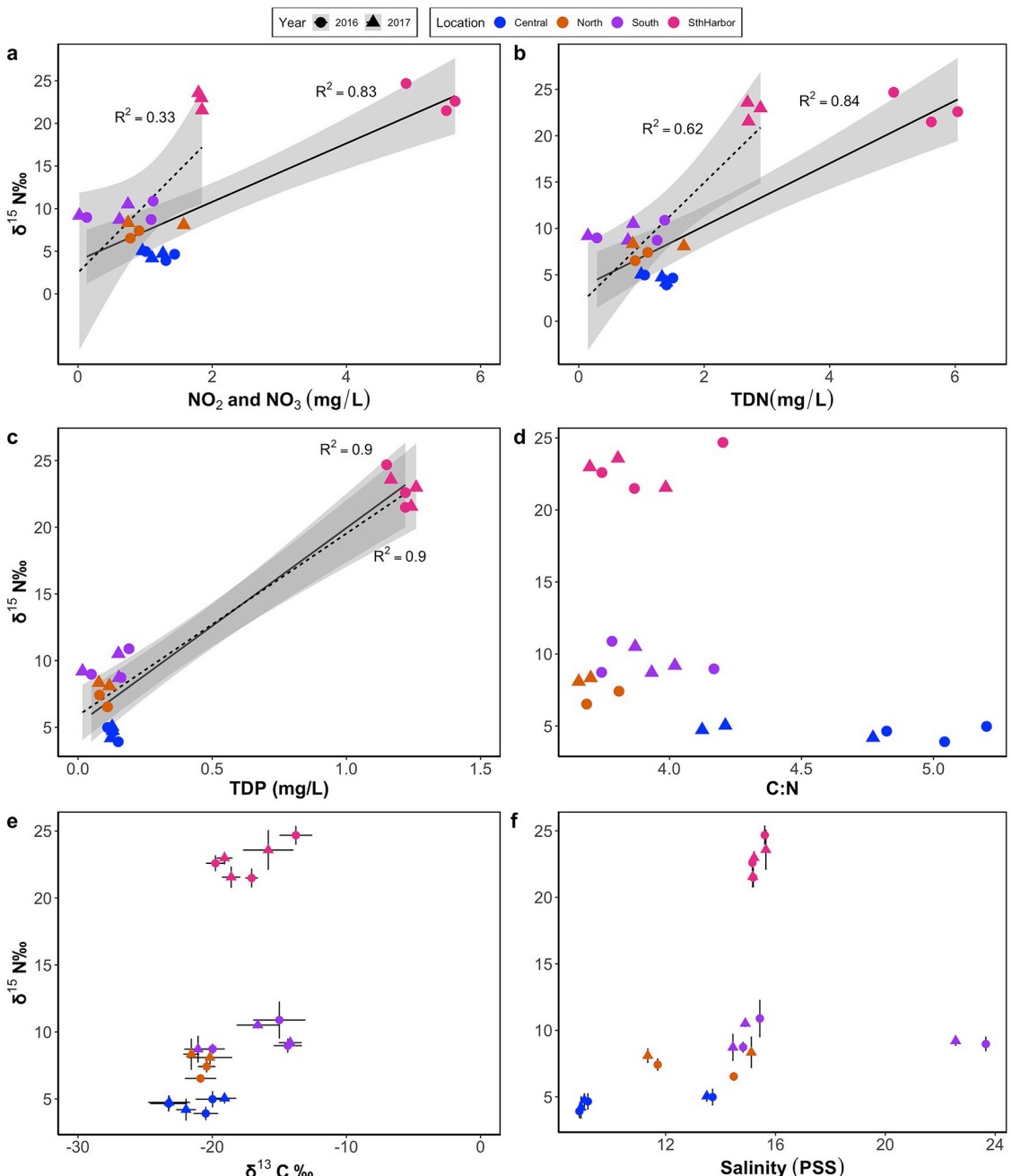

**Fig 3.** The relationship between the mean δ[15]N (‰) in *H. rubra* tissue to the mean concentration of (a) NO₃+NO₂ (mg/L), (b) TDN (mg/L), (c) TDP (mg/L), and (f) salinity (PSS) in water samples and the mean (d) C:N ratio and (e) δ [13]C (‰) in *H. rubra* tissues within 11 anchialine pools in Kaloko-Honokōhau National Historical Park. Colors represent pool cluster locations (North boundary, Central, South boundary, and South of Honokohau Harbor). Sampling occurred in 2016 (circles) and 2017 (triangles). Statistically significant linear regressions by year are indicated with 95% confidence intervals in gray (Table 1). Error bars in (e) and (f) indicate standard deviation.

Keal84 (the control treatment) changed very little from the original mean δ[15]N value (8.99 ± 0.25 ‰) over the course of the experiment. In the control, weekly δ[15]N values ranged between 8.49 ± 0.50 ‰ and 9.31 ± 0.46 ‰ (n = 3) over 77 days. Variability among weekly samples was higher in the shrimp with the diet shift compared to the control group. The average

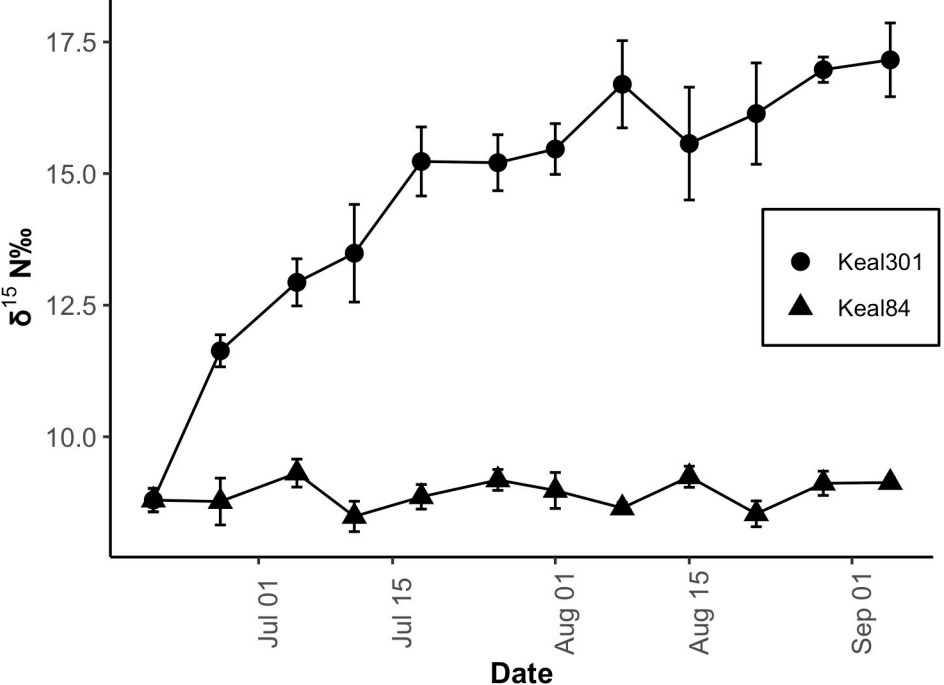

**Fig 4. Comparison of δ$^{15}$N (‰) in *H. rubra* tissues held in three replicate experimental microcosms with two different biofilm treatments: Control or wastewater influenced.** Closed circles represent shrimp that were fed biofilms from Keal84 (control) where shrimp were originally captured. Triangles represent shrimp fed biofilms from Keal301 where shrimp previously collected showed a wastewater signal indicated by elevated δ $^{15}$N levels (24‰). Each point represents the mean of three replicate samples. Error bars are the standard error of the mean.

dry weight of whole shrimp collected throughout the experiment was 5.5 ± 1.5 mg (n = 36) in the control treatment and 6.9 ± 2.3 mg (n = 35) in the experimental treatment.

Although isotopic equilibrium was not reached in shrimp fed the new diet over the 77 days of the experiment, the δ$^{15}$N values did approach an asymptote (Fig 5). The non-linear regression model describing changes in δ$^{15}$N of whole tissue as a function of time after the diet shift was calculated as δ$_t$ = 18–9.2 e$^{-0.034t}$ (r$^2$ = 42.08). The fractional turnover rate (λ) of nitrogen in whole animal tissue was estimated as 0.034. The time needed to achieve a 50% turnover (half-life, T$_{0.5}$) of nitrogen in the shrimp tissue was calculated as 20.4 days.

## Discussion

Our results show that nitrogen isotope levels within *H. rubra* tissue were a clear indicator of sewage in the groundwater flowing through some anchialine pools along the West Hawaiʻi coastal corridor. As expected, elevated levels of δ$^{15}$N within *H. rubra* whole tissue samples co-occurred with high levels of dissolved nitrogen and phosphorus. Furthermore, we found that *H. rubra* δ$^{15}$N samples showed low variability within individual pools and within pool complexes, reflecting the hydrologic connectivity in these areas. Finally, we found that nitrogen isotopes have a relatively long residence time in *H. rubra* tissues compared to groundwater in pools confirming that they provide a more time-integrative sample compared to water quality grab samples for dissolved nutrient levels.

At small spatial scales there was low variability of δ$^{15}$N measurements in shrimp tissues. Within-pool variability of individual shrimp δ$^{15}$N was low. Additionally, pools in proximity to

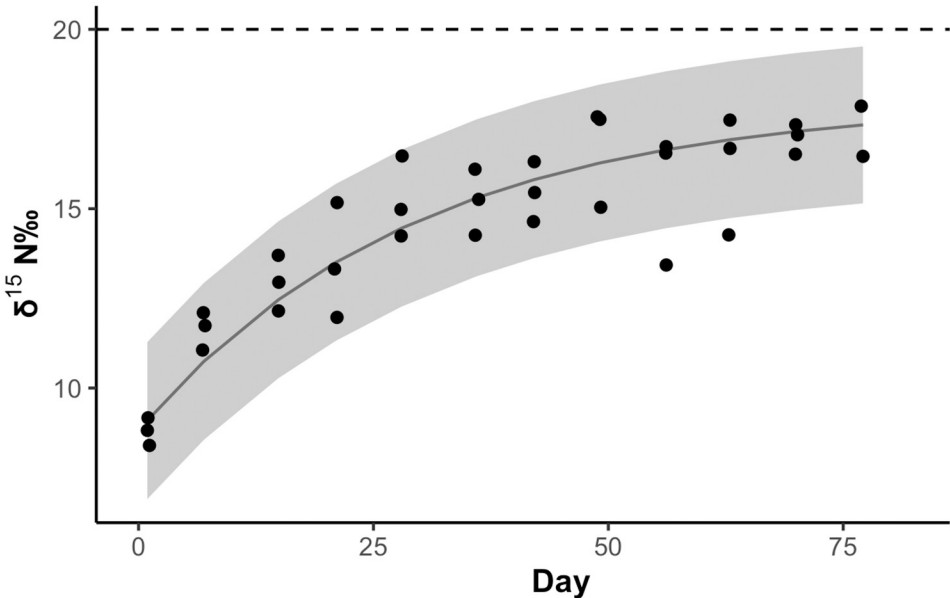

**Fig 5. *H. rubra* whole-tissue stable isotope ratios as a function of time after a diet shift.** A non-linear regression model (solid line) describes the changes in $\delta^{15}N$ of whole tissue as a function of time after a diet shift ($r^2$ = 42.08, $\delta_t$ = 18–9.2 $e^{-0.034t}$, Half-life $T_{0.5}$ = 20.4 days). The 95% confidence interval is shown in gray. The equation used is: $\delta_t = \delta_n + (\delta_o - \delta_n)e^{-(\lambda)t}$.

each other showed little difference in mean $\delta^{15}N$. Both within-pool and within-cluster similarity of $\delta^{15}N$ values indicate hydrologic connectivity between the pools in proximity. Although within-pool and within-pool cluster variability was low, $\delta^{15}N$ values did differ significantly between areas.

Nitrogen isotopes in *H. rubra* tissue confirmed that sewage inputs are entering anchialine pools via groundwater in the area south of the Honokohau Harbor. These pools are located between the ocean and the Kealakehe Wastewater disposal percolation basin. In these pools, average $\delta^{15}N$ levels in shrimp tissues were elevated (22.46‰ in 2015, 22.59‰ in 2016, and 23.06‰ in 2017). Prior to this study we measured similar $\delta^{15}N$ levels in *H. rubra* samples from this area (23.1‰ in 2013). In 2012, colleagues found equal $\delta^{15}N$ levels (23.1‰) in epilithon from pools in this same location [41]. These high levels are also consistent with measurements of $\delta^{15}N$ in dissolved nitrate from wastewater plumes downslope of the Kealakehe Wastewater Injection well and measurements in wastewater plumes on Maui [18]. Additionally, the elevated TDN, $NO_3+NO_2$, and TDP concentrations observed in these pools are consistent with previous measurements made in the Kealakehe Wastewater plume [18] and support the conclusion that wastewater contaminants are present in groundwater flowing through this pool complex.

Compared to the pools south of Honokohau Harbor, $\delta^{15}N$ levels were considerably lower elsewhere in the park. Wastewater may be entering the pools immediately north of Honokohau Harbor (especially Keal96 where levels were higher than 10‰). However, in the northern and central areas of the park, lower shrimp $\delta^{15}N$ values (< 8‰) coincided with lower nutrient concentration levels (TDN, $NO_3+NO_2$, and TDP) and indicated that little to no sewage influx was occurring in these areas. These results are supported by previous research in the park comparing dissolved nitrate $\delta^{15}N$, dissolved inorganic nutrients, and other indicators in groundwater observation wells and anchialine pools to background freshwater and saltwater levels

[18]. In this 2009 study, background dissolved nitrate $\delta^{15}$N levels were defined by a graphical saltwater-freshwater mixing line between a high elevation freshwater production well upslope of the park ($\delta^{15}$N = 2 ‰, presumed free of sewage influence) and a deep saline groundwater well adjacent the park ($\delta^{15}$N $\cong$ 3‰). Dissolved nitrate $\delta^{15}$N levels of 3‰ to 5‰ were recorded in the northern and central portions of the park. Although these values were slightly higher than the background groundwater levels, the researchers concluded that the isotopic enrichment in samples could reflect a fraction of septic nitrate from upslope development or some other unknown variation in nitrate isotopic composition from natural or human sources [18].

## Interpreting $\delta^{15}$N levels in primary consumers

In many studies that use $\delta^{15}$N levels to detect sewage in aquatic systems, algae species are used as bioindicators either alone or together with primary consumers [23, 27]. However, in many healthy anchialine pool habitats, macroalgae are absent and epilithion biomass may be extremely low. Conversely, *H. rubra* are ubiquitous in West Hawai'i and easily collected. For *H. rubra* to be considered a useful bioindicator, correct interpretation of $\delta^{15}$N levels in this grazing shrimp is necessary.

Compared to algae, which typically reflects the level of $\delta^{15}$N in the surrounding water body, consumer tissues have higher levels of $\delta^{15}$N due to trophic fractionation. Typically this trophic enrichment factor is estimated as +2‰ to 4‰ with each trophic level [30, 31]. Supporting previously published work, a study examining anchialine pool food web dynamics within ten pools in West Hawai'i, found that in many pools, *H. rubra* were +3‰ to 4‰ enriched compared to the primary producers available for grazing [41]. If the trophic enrichment value is assumed consistent, +3‰ to 4‰ could be added to the local baseline levels of dissolved $\delta^{15}$N in groundwater to estimate the expected $\delta^{15}$N for shrimp that are not impacted by sewage. Using the local groundwater baseline values of 2‰ to 3‰ [18], the expected $\delta^{15}$N levels in shrimp that are not impacted by sewage in West Hawai'i would be ~ 4‰ to 7‰.

Caveats apply to using a threshold level of nitrogen isotopes in primary consumer tissues as indication of sewage contamination. Nitrogen isotope results may be complicated by interaction of inputs from multiple sources, particularly fertilizer and sewage [13, 26, 59]. For example, if a groundwater body receives nitrogen inputs from agricultural fertilizer and sewage sources, the $\delta^{15}$N levels will be a mix of the two, and will be lower than the $\delta^{15}$N levels associated with sewage alone [13]. If fertilizer were the only input, lower $\delta^{15}$N levels may be observed along with high concentrations of TDN, $NO_3+NO_2$, and TDP.

Based on the positive linear relationships between shrimp-tissue $\delta^{15}$N and dissolved nutrients, there was no evidence of substantial fertilizer input in groundwater flowing through pools sampled in this study. This result is consistent with current land uses in the watershed of the park, which is primarily light industrial and high residential developments using OSDS upslope of the park (Fig 1b). In addition to collecting nutrient data, tracers such as Oxygen or Boron isotopes, over-the-counter medicines, or sewage-associated microbiota have been useful for identifying nitrogen sources entering groundwater and nearshore marine waters [13, 14, 18].

Tracing sewage contamination of groundwater in anchialine pool biota using $\delta^{15}$N assumes that the dominant nitrogen source entering the food web is from the groundwater. West Hawai'i is an arid landscape, and there is little surface runoff into anchialine pools [46]. Additionally, previous work in West Hawai'i showed that total dissolved nitrogen concentrations in anchialine pools did not depend on tree cover or introduced fish presence, and were dominated by Dissolved Inorganic Nutrients (DIN) flowing into the pools via groundwater [37].

Using 7 ‰ as the threshold for sewage, the $\delta^{15}$N levels in shrimp tissues collected from the pools south of Honokohau Harbor clearly indicate sewage inputs (22.46 to 23.06 ‰). In the

park to the north, average $\delta^{15}N$ levels observed within *H. rubra* tissues were 3.91‰ to 10.89‰ while the values for pools elsewhere in West Hawai'i ranged from 2.74‰ to 9.59‰. These results coincide with previously published work that pool averages for *H. rubra* $\delta^{15}N$ along the same geographic range varied from 3.6‰ to 12.6‰ [41]. Pools in the north and south boundary areas of the park may have some low levels of sewage input, the source of which is likely the high number of cesspools and septic systems farther upslope; a conclusion supported by previous results [18]. Additionally, sewage inputs may be occurring in pool Keah001 in the town of Kailua-Kona (9.59‰) and Lala01 (8.94‰), a pool located between a golf course and the Puako community where previous work showed significant sewage inputs from OSDS to groundwater flowing out to coral reefs [59]. Further monitoring and additional tracers could be used to confirm these conclusions.

### Nitrogen isotope turnover in *H. rubra*

The tissue turnover rate estimated for *H. rubra* by the mesocosm experiment supports earlier conclusions that the nitrogen composition of aquatic grazers provides a time-integrated view of nutrient sources in aquatic systems. Other studies have used epifaunal grazers in conjunction with primary producers to track sewage dispersal in aquatic environments because the slower tissue-turnover of consumers shows lower seasonal variability in their $\delta^{15}N$ values than primary producers or measures of dissolved nutrients [26, 60]. These conclusions are supported by results from our 2015 and 2016 field measurements that show tissue $\delta^{15}N$ was less variable between years than dissolved nitrogen (TDN and $NO_3+NO_2$) levels (Fig 3). Sampling stable isotopes of *H. rubra* or a similar grazer would be useful for detecting sewage inputs in unmonitored pools to identify target pools for monitoring, and periodically in some pools currently monitored by an infrequent grab-sample program.

Experimental isotopic diet-shift studies have been used to investigate the tissue turnover rate in several animal taxa. The rate that new isotopes are incorporated into tissue are a consequence of growth and tissue replacement. Therefore, life stage, metabolic processes and environment may all play a role in tissue turnover rates [56]. A review of these studies [39] shows that the half-life of carbon and nitrogen isotopes generally increases with animal body mass. However, the turnover rates are not consistently the same for carbon and nitrogen in the same organism. Furthermore, tissue-specific differences exist. For example, muscle and blood have longer half-lives compared to plasma and internal organs. In this study, the nitrogen half-life for whole-tissue *H. rubra* was estimated to be 20.4 days. In a study examining turnover in different tissues of the larger mantis shrimp, the average half-life of nitrogen in hemolymph was 20.0 ±5.7 days and in muscle was 50.4 ± 13.0 days [61]. Additionally, carbon had an eight-times more rapid turnover in hemolymph compared to nitrogen but was of the same magnitude in muscle [61]. *H. rubra* are much smaller than mantis shrimp; therefore, one would expect tissue turnover to be more rapid in *H. rubra*. Published estimates of nitrogen half-lives for small terrestrial arthropods in whole-tissue samples are between 1.5 for black fly larva to 9.3 days for springtail [39, 62, 63].

The tissue turnover time that we observed may have partially been a consequence of the mesocosm design. Although fresh biofilms were brought in from the sewage-influenced and control pools weekly, the water was not changed as frequently. Measurements of dissolved nutrients in mesocosm water showed consistent declines in $NO_3+NO_2$ between water changes (S3 Table). Therefore, shrimp may have quickly grazed down available food, and any new algal biomass produced in the mesocosms would not have been exposed to constant new supply of sewage influx to groundwater (and resulting elevated levels of $^{15}N$). The lack of water change may account for mesocosm shrimp reaching an asymptotic leveling off at lower $\delta^{15}N$ levels

($<$19‰) than was measured in shrimp tissues from pool Keal301 (22–23‰). Adding daily or weekly water changes to the weekly biofilm changes may have reproduced pool dynamics more closely and resulted in faster tissue turnover times as well as a higher asymptote.

Due to the small size of individual shrimp, we were not able to track growth of individuals prior to and during sampling [61]. Dry weight measurement of whole shrimp prior to stable isotope analysis indicated that the mass of shrimp in the control treatment remained relatively unchanged over the course of the experiment compared to the mass of shrimp fed biofilms from pool Keal301. Mean weight of the shrimp collected weekly from the sewage treatment (n = 3) increased over time compared to control shrimp (n = 3) (S1 Fig). For the shrimp exposed to sewage inputs (pool Keal301), this difference indicates that at least some of the change in $\delta^{15}$N over the course of the experiment was due to addition of new tissue and not just tissue turnover.

## Ecological effects of sewage contamination in pools

Hawaiian anchialine pools represent a unique habitat worthy of conservation. Numerous endemic, rare, and federally protected species rely on these habitats including the shrimp *Procaris hawaiana* and *Vetericaris chaceorum*, the orangeblack Hawaiian damselfly *Megalagrion xanthomelas*, and the Hawaiian Stilt *Himantopus mexicanus knudseni*, [42, 64]. Introduced fishes, changes in land-use that contribute to habitat loss, reductions or impairments of freshwater water quantity and quality, and rising sea levels threaten the integrity of pool ecosystems [43, 44, 65–67].

Sewage effluent has been shown to have a strong negative impact on numerous freshwater and marine ecosystems [2]. Multiple studies across a wide range of geographic locations have documented that anthropogenic nutrient enrichment can change the community structure and, in some cases, function of nearshore coastal ecosystems [1]. When elevated nutrient loads occur simultaneously with reduced grazing pressure, algal biomass increases have been documented in most fresh and marine ecosystems [68]. In some ecosystems such as rocky reefs, the combined effect of elevated nutrients and reduced grazing may be greater than either impact on its own, affecting algal biomass and community structure [68, 69]. Insights into how elevated nutrients and reduction in grazing may affect anchialine pools is evident in the cluster of pools south of Honokohau Harbor where pools with and without introduced fish are in proximity to each other. Despite consistently high dissolved nitrogen levels at all pools in this location over multiple years, pools without fish contain no sediment or macroalgae, contain high densities of *H. rubra* grazing on the epilithon on rocks (1000s per meter square) and contain other endemic shrimp. In nearby pools where fish are present, the biogenic sediment is several cm thick, *H. rubra* are rarely seen, and other endemic species are absent. Previous work in West Hawai'i shows that when introduced fish are present in anchialine pools, grazing by *H. rubra* is significantly reduced because shrimp are driven out of pools completely or may only enter pools at night [65, 66, 70]. Throughout West Hawai'i, pools without *H. rubra* are also more likely to have elevated Chl-a levels in water, contain biogenic sediment, and lack other endemic pool species [43]. The combination of reduced grazing from *H. rubra* and added nutrients from sewage appears to be causing high rates of primary production, sediment accumulation and pool degradation.

Future sea level change and the resulting change in groundwater flooding will change conditions in anchialine systems. Some pools will cease to be anchialine habitats as they become connected overland to the marine intertidal community. Other pools will enlarge and new habitat may form inland as low lying areas become flooded by groundwater lifted by a rising marine lens [67]. Increased groundwater flooding will allow introduced fishes to disperse into

new habitats in some locations. Finally, increased groundwater flooding due to sea level rise will increase the interaction between groundwater and sewage infrastructure and potentially increase contamination to pools [71].

From a management perspective, removal of introduced fishes has shown rapid recovery of *H. rubra* populations and should be pursued. In a healthy state, anchialine pool food webs are likely taking up some of the dissolved nutrients in groundwater prior to discharge onto near-shore coral reefs. Restoration work aimed at removing introduced fish and reducing sewage influx to groundwater will be beneficial to anchialine pools and potentially other coastal ecosystems that receive the groundwater after it flows through pools.

Wastewater contaminants that flow through groundwater-based habitats such as wetlands and anchialine pools will likely emerge in nearshore habitats via submarine groundwater discharge (SGD) [17]. Sewage effluent in SGD has been detected in multiple locations along the coast of the Hawaiian Islands [13, 18, 59, 72] and has been associated with coral disease [47], low species diversity [14], and macroalgal dominated systems [73]. On tropical reefs, elevated nutrient concentrations in conjunction with reduced herbivory are a probable mechanism for driving phase shifts from coral to algal dominance [73]. In many tropical locations where corals exist, anchialine systems should be considered an important resource for monitoring groundwater contamination prior to discharge in nearshore areas.

In addition to elevated nutrients, sewage effluent may also contain associated pathogens, pharmaceuticals, and other organic and inorganic toxins associated with wastewater [74]. Aside from nutrients, the effects of these contaminants in aquatic ecosystems, especially interactive effects with other contaminants, climate change or other stressors, are often poorly understood [3]. Published research on the effects of emerging contaminants of concern on anchialine pool biota is lacking.

## Conclusion

The results from this study suggest *H. rubra* tissue $\delta^{15}$N levels are a useful indicator of sewage in coastal groundwater ecosystems on the island of Hawai'i. In healthy anchialine systems, macroalgae are typically not available for this purpose. The primary consumer *H. rubra* is commonly found in pools in abundance, is easy to collect, and shows low variability in $\delta^{15}$N levels both within a single pool and between pools that are in proximity. Like other bioindicators, *H. rubra* tissues have relatively long-turnover rates making them a more time-integrated sample compared to water quality measurements of dissolved nutrient levels. Although *H. rubra* are endemic to Hawai'i, primary consumers found in anchialine systems elsewhere in the world may also work well in determining sewage contamination in groundwater-dependent ecosystems. Used in conjunction with traditional water quality monitoring methods as well as other tracers, *H. rubra* or other primary consumers may be helpful to detect sewage contamination in groundwater prior to discharge and dilution in nearshore marine waters.

## Supporting information

**S1 Fig. Average dry weight of shrimp (n = 3) collected weekly from mesocosms prior to δ** **$^{15}$N analysis.** Error bars are the standard deviation. Pool Keal84 is the control treatment and pool Keal301 is the sewage influenced pool.
(TIF)

**S1 Table. Anchialine pools where *H. rubra* were collected for δ $^{15}$N analysis between 2015 to 2017.** Location is the island-wide naming convention for pools ('Island_land

division_number'). ID is a shortened version used in this paper. Year of sampling is indicated by an x.
(DOCX)

**S2 Table. Water quality characteristics and nutrients for 11 anchialine pools in and south of Kaloko-Honokohau National Historical Park during 2016 and 2017.** Stable isotopes were from shrimp tissues with the pools sampled. Standard deviation is shown with the mean (n = 3) for stable isotopes.
(DOCX)

**S3 Table. Mean values of water quality measures in mesocosm containers (mean ± standard deviation; n = 3) at the start of the experiment, at day 50, at day 50 after the new water was placed in the mesocosm, and at day 77 at the end of the experiment.** Water for the sewage exposed pool treatments (301) was collected at that pool. Pool for the control treatment was collected at pool 84.
(DOCX)

**S1 Appendix. Descriptions and sources for raw data tables of all water quality analysis and stable isotope analysis results for individual samples.** Files are available on the NPS- IRMA website (https://doi.org/10.57830/2299671).
(DOCX)

## Acknowledgments

We thank Kaileʻa Annandale and Ashley Pugh for their assistance with the mesocosm study and water quality sampling.

## Author Contributions

**Conceptualization:** Lisa C. Marrack.

**Data curation:** Lisa C. Marrack.

**Formal analysis:** Lisa C. Marrack.

**Funding acquisition:** Lisa C. Marrack, Sallie C. Beavers.

**Investigation:** Lisa C. Marrack, Sallie C. Beavers.

**Methodology:** Lisa C. Marrack, Sallie C. Beavers.

**Project administration:** Lisa C. Marrack.

**Resources:** Lisa C. Marrack, Sallie C. Beavers.

**Visualization:** Lisa C. Marrack.

**Writing – original draft:** Lisa C. Marrack.

**Writing – review & editing:** Lisa C. Marrack, Sallie C. Beavers.

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
