## [Decision Letter · Decision Letter 0]

29 May 2023

PONE-D-23-04906Anchialine pool shrimp (Halocaridina rubra) as an indicator of sewage in coastal groundwater ecosystems on the island of HawaiʻiPLOS ONE

Dear Dr. Marrack,

Thank you for submitting your manuscript to PLOS ONE. After careful consideration, we feel that it has merit but does not fully meet PLOS ONE’s publication criteria as it currently stands. Therefore, we invite you to submit a revised version of the manuscript that addresses the points raised during the review process.

We look forward to receiving your revised manuscript.

Kind regards,

Malik Muhammad Akhtar, PhD, Postdoc

Academic Editor

PLOS ONE

Additional Editor Comments:

In Abstract: Replace “ranged between 2.74 and 58 22.46‰.” with “ranged between 2.74‰ and 58 22.46‰.”

Used uniform symbols of “δ15N.”

Include value “The significant positive relationship between δ15N and dissolved nitrogen and phosphorus”

Include methodology (sample testing technique and data analysis) and a key recommendation in Abstract.

Include/replace references of recent studies of last three years in introduction and discussions part.

The start of “Conclusions part” with suggestions is not a good idea. Give conclusions of your study avoiding references

Add coordinates in figure-1

Use uniform unit for scale bar in figure-1

Reviewers' comments:

Reviewer's Responses to Questions

**Comments to the Author**

1. Is the manuscript technically sound, and do the data support the conclusions?

Reviewer #1: Yes

Reviewer #2: Yes

2. Has the statistical analysis been performed appropriately and rigorously? 

Reviewer #1: Yes

Reviewer #2: Yes

3. Have the authors made all data underlying the findings in their manuscript fully available?

Reviewer #1: Yes

Reviewer #2: Yes

4. Is the manuscript presented in an intelligible fashion and written in standard English?

Reviewer #1: Yes

Reviewer #2: Yes

5. Review Comments to the Author

Reviewer #1: The study founds that the nitrogen half-life in H. rubra tissue was estimated to be 20.4 days, indicating that this species could provide a time-integrative sample for detecting sewage in groundwater. This method could be used in conjunction with standard monitoring methods for more accurate detection, especially in anchialine habitats that typically do not support macroalgae. The study highlights the potential risks to marine life and human health from wastewater and other pollutants entering coastal ecosystems via groundwater, and the importance of monitoring and mitigating these risks.

this is considerable addition in the existing knowledge

Reviewer #2: The manuscript is very impressive and covers all data including statistical evaluations in detail. However a minor addition of a paragraph highlighting how quality assurance was undertaken will be appreciated. One or two very recent (2021/2022) relevant citations in the introduction part will make the manuscript amazing.

6. PLOS authors have the option to publish the peer review history of their article (what does this mean?). If published, this will include your full peer review and any attached files.

Reviewer #1: **Yes: **Muhammad Aslam

Reviewer #2: **Yes: **Sonia Tariq

---

## [Author Response · Author response to Decision Letter 0]

7 Jul 2023

Dear Reviewers:

We greatly appreciate your time reviewing the manuscript. Thank you for the suggestions. We have corrected and addressed all comments. The response letter that we uploaded highlights each suggestion and the associated corrections /modifications to the manuscript.

Best regards,

Lisa Marrack

---

## [Decision Letter · Decision Letter 1]

13 Aug 2023

Anchialine pool shrimp (Halocaridina rubra) as an indicator of sewage in coastal groundwater ecosystems on the island of Hawaiʻi

PONE-D-23-04906R1

Dear Dr. Marrack,

We’re pleased to inform you that your manuscript has been judged scientifically suitable for publication and will be formally accepted for publication once it meets all outstanding technical requirements.

Kind regards,

Malik Muhammad Akhtar, PhD, Postdoc

Academic Editor

PLOS ONE

Additional Editor Comments (optional):

Reviewers' comments:

Reviewer's Responses to Questions

**Comments to the Author**

1. If the authors have adequately addressed your comments raised in a previous round of review and you feel that this manuscript is now acceptable for publication, you may indicate that here to bypass the “Comments to the Author” section, enter your conflict of interest statement in the “Confidential to Editor” section, and submit your "Accept" recommendation.

Reviewer #1: All comments have been addressed

Reviewer #2: All comments have been addressed

2. Is the manuscript technically sound, and do the data support the conclusions?

Reviewer #1: Yes

Reviewer #2: Yes

3. Has the statistical analysis been performed appropriately and rigorously? 

Reviewer #1: Yes

Reviewer #2: Yes

4. Have the authors made all data underlying the findings in their manuscript fully available?

Reviewer #1: Yes

Reviewer #2: Yes

5. Is the manuscript presented in an intelligible fashion and written in standard English?

Reviewer #1: Yes

Reviewer #2: Yes

6. Review Comments to the Author

Reviewer #1: Author has addressed all required questions and made all modification suggested for the improvement of manuscript

Reviewer #2: (No Response)

7. PLOS authors have the option to publish the peer review history of their article (what does this mean?). If published, this will include your full peer review and any attached files.

Reviewer #1: **Yes: **Muhammad Aslam, Lasbela University of Agriculture Water and Marine Sciences, Pakistan

Reviewer #2: **Yes: **Sonia Tariq

---

## [Editor Report · Acceptance letter]

21 Aug 2023

PONE-D-23-04906R1 

Anchialine pool shrimp (*Halocaridina rubra*) as an indicator of sewage in coastal groundwater ecosystems on the island of Hawaiʻi 

Dear Dr. Marrack:

I'm pleased to inform you that your manuscript has been deemed suitable for publication in PLOS ONE. Congratulations! Your manuscript is now with our production department. 

Kind regards, 

on behalf of

Professor Malik Muhammad Akhtar 

Academic Editor

PLOS ONE